# Effects of LiBF_4_ Addition on the Lithium-Ion Conductivity of LiBH_4_

**DOI:** 10.3390/molecules27072187

**Published:** 2022-03-28

**Authors:** Laura M. de Kort, Valerio Gulino, Didier Blanchard, Peter Ngene

**Affiliations:** 1Materials Chemistry and Catalysis, Department of Chemistry, Debye Institute for Nanomaterials Science, Utrecht University, Universiteitsweg 99, 3584 CS Utrecht, The Netherlands; l.m.dekort@uu.nl (L.M.d.K.); v.gulino@uu.nl (V.G.); 2The European Synchrotron Facility (ESRF), 71 Avenue des Martyrs, 38000 Grenoble, France; didier.blanchard@esrf.fr

**Keywords:** solid-state electrolytes, complex hydrides, lithium borohydride, ionic conductivity, ion substitution, interface effects

## Abstract

Complex hydrides, such as LiBH_4_, are a promising class of ion conductors for all-solid-state batteries, but their application is constrained by low ion mobility at room temperature. Mixing with halides or complex hydride anions, i.e., other complex hydrides, is an effective approach to improving the ionic conductivity. In the present study, we report on the reaction of LiBH_4_ with LiBF_4_, resulting in the formation of conductive composites consisting of LiBH_4_, LiF and lithium *closo*-borates. It is believed that the in-situ formation of *closo*-borate related species gives rise to highly conductive interfaces in the decomposed LiBH_4_ matrix. As a result, the ionic conductivity is improved by orders of magnitude with respect to the Li-ion conductivity of the LiBH_4_, up to 0.9 × 10^−5^ S cm^−1^ at 30 °C. The insights gained in this work show that the incorporation of a second compound is a versatile method to improve the ionic conductivity of complex metal hydrides, opening novel synthesis pathways not limited to conventional substituents.

## 1. Introduction

In our current society, rechargeable lithium-ion batteries are crucial energy storage devices for a wide variety of applications, ranging from small portable electronics to electric vehicles. The performance and safety requirements for automotive electrification and grid-energy storage systems, however, cannot be met by the commercially available batteries [1,2]. Hence, the development of batteries with higher energy densities and improved safety is crucial. The performance limitations in commercial Li-ion batteries are caused by their organic liquid electrolytes that often suffer from poor electrochemical and thermal stability, which poses safety risks and limits storage capacity [3]. Replacement of the liquid electrolyte with a solid electrolyte could overcome these persistent problems. Solid-state electrolytes (SSEs) are inherently safer and, in many cases, compatible with high energy density electrodes, such as metallic lithium [4]. Consequently, the realization of all-solid-state batteries implementing SSEs can lead to safer batteries that store more energy.

The anticipated benefits of SSEs has led to a growing number of studies on inorganic lithium ion conductors [5], such as garnet-type [6,7], perovskite-type [8,9] and sulphide-type materials [10,11] The research scope on potential Li-ion conductors was recently extended to complex metal hydrides following the surprising discovery of fast Li-ion mobility in LiBH_4_ [12]. Complex metal hydrides exhibit several unique material properties that are interesting for battery application, i.e., low molecular weight, easy deformability and high compressibility, and compatibility with a lithium metal anode [13,14,15,16]. Among others, these benefits have spurred research on complex metal hydrides as promising SSEs for all-solid-state Li-ion batteries.

A main disadvantage for metal hydrides, and many other solid electrolytes, is their low room temperature (RT) ionic conductivity. Complex hydrides typically exhibit high ionic conductivity as a result of a reversible structural phase change that requires high temperatures [12,17,18]. For example, fast lithium-ion mobility and high conductivity (~10^−3^ S cm^−1^) in LiBH_4_ is only observed after a phase transition from the orthorhombic to the hexagonal phase at 110 °C [12]. Therefore, the development of strategies that enhance RT Li-ion conductivity is of major importance for the application of metal hydride-based ion conductors.

From various methods developed to boost ionic conductivity in complex hydrides, e.g., nanoconfinement and interface engineering [19,20,21,22], partial ionic substitution has proven to be particularly useful [23,24,25,26,27,28]. Previous reports reveal that the incorporation of lithium halides (LiX, with X = Br^−^, I^−^) or lithium amide (LiNH_2_) in the LiBH_4_ crystal structure leads to significant improvement in ionic transport. Depending on the substituting anion, partial anionic substitution of BH_4_^−^ either leads to stabilization of the conductive hexagonal phase at lower temperatures (*h*-Li(BH_4_)_1−y_(X)_y_) [15,29,30] or to the formation of a new highly conductive crystal structure, e.g., Li_2_(BH_4_)(NH_2_) [27,28].

The use of anion substitution for conductivity enhancement in LiBH_4_ has so far been limited to the addition of halides and other complex hydrides. A complex anion that could also be interesting as a partial substituent for [BH_4_]^−^ is the [BF_4_]^−^ anion. LiBF_4_ is a well-known Li-ion conductor for liquid electrolytes, but it has not been investigated as SSE because it exhibits very poor conductivity as a solid electrolyte. Partial ionic substitution of BF_4_^−^ in LiBH_4_ has been investigated as a method to improve the hydrogen storage properties of metal borohydrides [31,32,33,34,35]. Different from LiX and LiNH_2_, in this case, it was reported that F^−^ is exchanged for H^−^, forming a [BH_4−x_F_x_]^−^ complex. Theoretical studies proposed that intramolecular hydrogen-fluorine exchange in MBH_4_-MBF_4_ systems (M = Li, Na, K) significantly alters the borohydride structure [31,32,33]. It has been speculated that the strong electronegativity of the fluoride atom weakens the force of attraction between the B and the H in the complex anion [36]. Similarly, electrostatic interactions between the [BH_4−x_F_x_]^−^ anion with the Li^+^ cation could be attenuated, thereby improving Li-ion mobility. The predicted effects in the LiBH_4_-LiBF_4_ system for hydrogen storage have been experimentally reported by Richter et al. [36]. However, it was demonstrated that the formation of [BH_4−x_F_x_]^−^ causes a strong destabilization of LiBH_4_, leading to a decomposition pathway forming diborane gas and solid decomposition products such as LiF and Li_2_B_12_H_12_. This decomposition reaction pathway led the authors to conclude that this system does not hold promise of any practical usage in hydrogen desorption/absorption cycling [36]. Even if the conclusion of the authors can be considered correct, the studied system could be of interest in the SSE research field. In fact, it has been suggested that *closo*-borate anions (e.g., B_12_H_12_^2−^) have lower reduction ability than BH_4_^−^ and thus show higher electrochemical stability, in addition to a higher RT Li-ion conductivity [37].

Recently, Zhu et al. [36] reported that partial dehydrogenation of LiBH_4_ results in the in-situ formation of highly conductive Li−B−H complexes, containing LiBH_4_, [Li_2_B_12_H_11+1/n_]_n_ and LiH. A dramatic increase in Li-ion conductivity to 2.7 × 10^−^^4^ S cm^−^^1^ at 35 °C is observed, likely due to the presence of a highly conductive interface between [Li_2_B_12_H_11+1/n_]_n_ and LiBH_4_ [38]. Although the exact phase is still under debate, these Li−B−H complexes are believed to contain *closo*-borates [39]. We expect that similar in-situ formed Li−B−H complexes can be derived from the destabilization of LiBH_4_ caused by LiBF_4_, which may exhibit superior Li-ion conducting performances.

In this work, we explore the impact of LiBF_4_ incorporation on LiBH_4,_ especially looking into the ionic conductivity of in-situ formed Li-B-H complexes. The structural properties and ionic conductivity of the synthetized mixtures are analysed to investigate how the presence of BF_4_^−^ in the mixtures affects the Li-B-H complex composition and structure, as well as the lithium-ion mobility.

## 2. Experimental Section

### 2.1. Solid-State Synthesis

Several LiBH_4_-LiBF_4_ and LiBH_4_-Li_2_B_12_H_12_ mixtures were prepared by physically mixing LiBH_4_ (purity > 95%, Sigma-Aldrich, St. Louis, MO, USA) with either 20 mol% of LiBF_4_ (purity > 98%, Sigma-Aldrich) or 25 mol% of anhydrous Li_2_B_12_H_12_ (purity > 98%, Katchem, Prague, Czech Republic) in an agate mortar approximately 5 min. The physical mixtures were transferred into a stainless-steel reactor, which was placed in a stainless-steel high-pressure autoclave (Parr, 250 mL). After pressurizing the autoclave with 50 bar of H_2_, the physical mixtures were heated to 150 °C or 280 °C (heating rate 2.5 °C min^−1^) for 30 min. The samples were collected from the autoclave after the heat treatment and analysed further. An overview of all prepared samples is provided in Appendix A. All storage and handling of the chemicals and prepared samples was carried out in an argon-filled glovebox (H_2_O and O_2_ < 0.1 ppm).

### 2.2. Structural Characterization

X-ray powder diffraction (XRD) analysis was performed with a Bruker-AXS D8 Advance X-ray diffractometer (Co Kα_1_ = 1.78897 Å). The samples were placed in an airtight sample holder, preventing air and moisture exposure. Diffractograms were recorded at RT from 20° to 80° 2θ range.

Diffuse reflectance infrared Fourier transform spectroscopy (DRIFTS) measurements were performed on a Perkin-Elmer 2000 spectrometer equipped with an MCT detector. Using an airtight sample holder with KBr windows, samples were measured without any air or moisture exposure. Spectra were acquired from 900 cm^−1^ to 4500 cm^−1^ with a resolution of 4 cm^−1^.

### 2.3. Electrochemical Impedance Spectroscopy

Electrochemical impedance spectroscopy (EIS) measurements were performed using a Princeton Applied Research Parstat 2273 connected to a custom-made measurement cell in a Büchi B-585 glass oven. Two stainless steel cylinders with Li foil (Sigma-Aldrich, purity 99.9%, 0.38 mm thick) were placed on one side and configured as electrodes. Using a standard 13 mm pellet press, an approximately 80–150 mg sample was pressed between the electrodes with a pressure of approximately 1.5 ton cm^−2^. During a typical conductivity measurement, the measurement cell was incrementally heated to 130 °C (∆T = 10 °C) and incrementally cooled to room temperature (∆T = 20 °C). At each increment the temperature was allowed to equilibrate before an EIS measurement was performed using a 10 mV of amplitude and with frequencies from 1 MHz to 1 Hz.

### 2.4. Linear Sweep Voltammetry

Linear sweep voltammetry (LSV) was used to determine the oxidative stability of the LiBH_4_-20% LiBF_4_ sample. The sample was mixed with carbon black (CB, Ketjenblack EC600JD) in a weight ratio of 90:10 using an agate mortar. A pellet was obtained by pressing approximately 5 mg of electrolyte-carbon mixture and 50 mg of the electrolyte at 1.5 ton cm^−2^. After compressing, a cell was formed with stainless-steel as the working electrode in contact with the electrolyte-carbon mixture and a lithium disk as the counter electrode in contact with the solid electrolyte. The linear sweep measurement was performed at 60 °C with a voltage range from 1.0 V to 5.0 V versus Li^+^/Li at a scanning rate of 0.1 mVs^−1^.

## 3. Results and Discussion

### 3.1. Structural Characterization and Conductivity of LiBH_4_-LiBF_4_ Mixtures

The impact of the addition of LiBF_4_ on the crystal structure of the Li-B-H complexes has been determined using XRD. In Figure 1a, the diffractograms of a mixture consisting of 80 mol% of LiBH_4_ and 20 mol% LiBF_4_ prepared via solid-state reaction at 150 °C and 280 °C are shown. The XRD patterns of LiBH_4_ and LiBF_4_ are provided for comparison. In the diffraction pattern of the LiBH_4_-LiBF_4_ mixture heated at 150 °C, the main diffraction peaks characteristic of both starting compounds are still observed. Additionally, two new peaks have appeared at 45 and 53 2θ°, that correspond to cubic (*Fm-3m*) LiF. This can be explained considering that the decomposition of LiBH_4_ in the presence of LiBF_4_ starts at 80 °C and LiF has been reported as a decomposition product [36]. During the reaction, hydrogen-fluorine exchange results in the formation of LiBH_4−x_F_x_ [36], which is not stable and decomposes to form LiF.

Similarly, the diffraction pattern of the LiBH_4_-LiBF_4_ mixture prepared at 280 °C predominantly contains peaks related to LiF; however, in this case no clear diffraction peaks of the starting compounds can be identified. Thus, at a reaction temperature of 280 °C, the decomposition reaction proceeds further in agreement with results reported by Richter et al. [36] Note that with XRD only crystalline compounds can be identified. Therefore, it is not clear whether LiF is the only solid decomposition product or if an unidentified amorphous phase is formed as well. For example, Zhu et al. reported that highly conductive amorphous Li-B-H complexes can form during the LiBH_4_ decomposition [38].

Because with the XRD analysis the presence of possible amorphous Li-B-H complexes could not be confirmed, the structural properties of the LiBH_4_-LiBF_4_ mixtures were further analysed using DRIFTS measurements. This technique is not limited to crystalline compounds, which means that it can be used to identify the composition of the crystalline and amorphous phases present in the LiBH_4_-LiBF_4_ mixtures. The DRIFTS absorbance spectra obtained for LiBH_4_-LiBF_4_ synthesized at 150 °C and 280 °C are shown in Figure 1b, as well as the spectra for pristine LiBH_4_ and LiBF_4_. Pristine LiBH_4_ typically shows several characteristic absorption bands. The most predominant bands, corresponding to [BH_4_^−^] stretching vibrations, can be seen between 2000 and 2800 cm^−1^. Other characteristic LiBH_4_ vibrations can be seen between 1000 and 1300 cm^−1^ ([BH_4_^−^] bending) and 3200 and 3700 cm^−1^ ([OH] stretching from adsorbed moisture) [40,41,42]. Though less intense, these characteristic bands are seen in the spectra of both LiBH_4_-LiBF_4_ mixtures, indicating that LiBH_4_ is still present in both mixed compounds, hence the starting compounds have not completely decomposed. Note that the bands related to LiBH_4_ vibrations are especially pronounced in the DRIFTS spectrum of LiBH_4_-LiBF_4_ prepared at 280 °C, even though no peaks related to crystalline LiBH_4_ were present in the diffractogram, suggesting that residual amorphous LiBH_4_ is present in the compound.

Besides the LiBH_4_ vibrations, multiple unidentified peaks are present in both LiBH_4_-LiBF_4_ spectra, specifically around 2500 cm^−1^ and at 1800 cm^−1^. Surprisingly, none of these peaks is related to the presence of LiBF_4_ or LiF (Figure 1b) [40,43]. Clearly, decomposition of the unstable LiBH_4−x_F_x_ complex does not only result in the formation of LiF, but also in an unidentified amorphous compound. Similar to the Li-B-H complexes obtained during LiBH_4_ decomposition described by Zhu et al., this unidentified compound could be beneficial for the overall conductivity of the LiBH_4_-LiBF_4_ compounds [38]. Therefore, the nature of this compound has been studied in more detail, which will be discussed later in this work.

The impact of the decomposition reaction on Li-ion conductivity in the prepared LiBH_4_-LiBF_4_ mixed compounds is determined using electrochemical impedance spectroscopy (EIS). The obtained Nyquist plots display single semicircles that could be analysed with a simple one-phase model, as is shown in Appendix A. Additionally, exemplary Bode plots corresponding to the data are shown in Appendix A. In this way, the resistance and corresponding conductivity are determined over a temperature range from room temperature to 130 °C. In Figure 1c and Appendix A, the temperature dependence of the ionic conductivity of the LiBH_4_-LiBF_4_ mixtures prepared at 150 °C and 280 °C is displayed using an Arrhenius plot. Comparison to the ionic conductivity of LiBH_4_ shows that ion transport in the LiBH_4_-LiBF_4_ mixtures prepared at 280 °C has improved at temperatures lower than 100 °C, reaching 0.9 × 10^−5^ S cm^−1^ at 30 °C, which is three orders of magnitude higher than that of LiBH_4_. In contrast, the mixture prepared with a lower reaction temperature exhibits a conductivity lower than that of LiBH_4_ over the entire temperature range.

At temperatures above 110 °C, the same temperature-dependent Li-ion conductivity behaviour as LiBH_4_ has been measured for the mixture treated at 150 °C, i.e., a drastic increase of the Li-ion conductivity. This behaviour indicates that the sample contains unreacted LiBH_4_ that undergoes a phase transition from the orthorhombic to the hexagonal phase, increasing the Li-ion conductivity. These results agree with both the diffraction and IR data reported in Figure 1. The lower ion conductivity with respect to pure LiBH_4_ can be explained considering that the sample also contains non-conductive phases, i.e., unreacted LiBF_4_ and LiF. The LiBH_4_-LiBF_4_ mixture prepared at 280 °C also exhibits a conductivity enhancement above 110 °C related to the phase transition of LiBH_4_, though it is not as pronounced. The conductivity above 110 °C is lower than that of pristine LiBH_4_, which again can be attributed to the presence of non-conductive LiF.

The difference in conductivity between the mixtures prepared with different reaction temperature (see also Appendix A) indicates that the reaction temperature of the prepared LiBH_4_-LiBF_4_ system plays an important role. It is not likely that LiF predominantly contributes to the enhanced conductivity, as it does not exhibit high room temperature conductivity by itself, typically below 10^−7^ S cm^−1^ [44,45]. On the other hand, the incorporation of LiF could lead to the formation of a LiF-substituted LiBH_4_ solid solution in which the hexagonal LiBH_4_ phase is stabilized at lower temperatures. However, previous studies on halide substituted LiBH_4_ demonstrated that the RT stability range of the hexagonal phase decreases with the anionic radius. In fact, it has been reported that h-Li(BH_4_)_1−α_(I)_α_ solid solutions are stable at RT in the range of 0.18 ≤ α ≤ 0.50, [46,47] h-Li(BH_4_)_1−α_(Br)_α_ in the range 0.3 ≤ α ≤ 0.55 [29], while Cl^−^ is not able to stabilize the high temperature phase at RT [46]. The ionic size of F^−^ is considerably smaller with respect to BH_4_^−^ as well as to the other halides, i.e., *r*(F^−^) = 1.33 Å, *r*(Cl^−^) = 1.81 Å, *r*(Br^−^) = 1.96 Å, *r*(I^−^) = 2.20 Å and *r*(BH_4_^−^) = 2.03 Å [48], which makes stabilization of the hexagonal phase at ambient temperature unlikely. This hypothesis is further confirmed by the absence of reflections related to hexagonal LiBH_4_ in the LiBH_4_-LiBF_4_ diffraction patterns (Figure 1a). Consequently, it is not expected that the formation of LiF contributes to conductivity enhancement in the LiBH_4_-LiBF_4_ compound. 

Nevertheless, it is clear that a highly conductive compound forms after the solid-state reaction occurring at 280 °C, during which hydrogen-fluoride substitution and subsequent decomposition of LiBH_4−x_F_x_ occur. Because an improved conductivity is only seen in this case, it seems that the decomposition reaction that results in the presence of amorphous LiBH_4_ and the formation of the unidentified compound (only observed in DRIFTS) is of vital importance to the conductivity enhancement. Richter et al. established a decomposition pathway of LiBH_3_F (Figure 2) that could explain the conductivity enhancement in the LiBH_4_-LiBF_4_ mixture [36].

Their analysis illustrates that the formation of LiBH_3_F from hydrogen-fluorine substitution does not solely result in the formation of LiF. The highly unstable LiBH_3_F compound decomposes to form both LiF and diborane (B_2_H_6_). Finally, diborane can react with an excess of LiBH_4_ to form Li_2_B_12_H_12_. Because the studied composition consists of 80 mol% of LiBH_4_, it is possible for this final reaction (Figure 2c) to occur. This decomposition pathway can support our observations on the effect of hydrogen-fluorine exchange and decomposition of the LiBH_4_-LiBF_4_ mixtures together with the enhancement of the Li-ion conductivity.

### 3.2. LiBH_4_-Li_2_B_12_H_12_ Composites 

To investigate if the high ionic conductivity is related to the formation of Li_2_B_12_H_12_-like or LiBH_4_-Li_2_B_12_H_12_ composites during the decomposition reaction, mixtures of LiBH_4_ with 10 to 75 mol% Li_2_B_12_H_12_ (see Appendix A) were prepared following the synthetic procedure used for the mixture LiBH_4_-LiBF_4_ synthetized at 280 °C. The composition based on 25 mol% Li_2_B_12_H_12_ will be discussed below because it resembles the composition of the LiBH_4_-LiBF_4_ mixture. The DRIFTS and XRD spectra of the LiBH_4_-25% Li_2_B_12_H_12_ sample prepared are shown in Figure 3, together with those obtained from the LiBH_4_-LiBF_4_ system, while in Appendix A the other LiBH_4_-Li_2_B_12_H_12_ samples are presented. In Figure 3a, the DRIFTS spectra of both LiBH_4_-LiBF_4_ and LiBH_4_-Li_2_B_12_H_12_ contain strong absorption bands at 2535 cm^−1^ and 2480 cm^−1^. The previously unidentified peaks in the LiBH_4_-LiBF_4_ spectrum can thus be associated with the presence of Li_2_B_12_H_12_, which displays characteristic [B-H] stretching vibrations exactly at these wavenumbers [49,50,51]. Other similarities in the DRIFTS spectra of LiBH_4_-LiBF_4_ and LiBH_4_-Li_2_B_12_H_12_ are also observed. The comparison reveals that, in addition to the pronounced bands around 2500 cm^−1^, smaller peaks between 3700 and 3000 cm^−1^, at 1835 cm^−1^ and at 1015 cm^−1^, can also be explained by the presence of Li_2_B_12_H_12_ [49,50,51]. Together with the XRD results discussed previously (Figure 1b), this illustrates that the prepared LiBH_4_-LiBF_4_ mixtures decompose to form crystalline LiF and likely amorphous *closo*-borate-related compounds with residual LiBH_4_. 

This assertion is further investigated by comparing the XRD patterns of LiBH_4_-LiBF_4_ to that of LiBH_4_-Li_2_B_12_H_12_ shown in Figure 3b. Notably, the pattern collected for the LiBH_4_-Li_2_B_12_H_12_ contains characteristic diffraction peaks related to both starting compounds, which clearly demonstrates that both phases are still present and crystalline. This is different from the pattern of LiBH_4_-LiBF_4_, where, besides LiF, no other crystalline phase is detected. Note that a small peak indicative of Li_2_B_12_H_12_ can be observed at 24 2θ° in the LiBH_4_-LiBF_4_ mixture pattern, strengthening the possibility that Li_2_B_12_H_12_ has formed. However, in general it seems that the reaction between LiBF_4_ and LiBH_4_ leads to partial decomposition forming crystalline LiF and amorphous Li_2_B_12_H_12_. Several studies have shown that the incorporation of Li_2_B_12_H_12_ in the LiBH_4_ matrix could lead to enhanced lithium-ion mobility [38,52,53,54]. Similar to the Li-B-H compound proposed by Zhu et al. [38], after reaction the LiBH_4_-LiBF_4_ mixture consists of residual LiBH_4_ in contact with LiF and amorphous Li_2_B_12_H_12_. Accordingly, the conductivity enhancement could originate from the in-situ formation of Li_2_B_12_H_12_. 

To further investigate this hypothesis, electrochemical impedance spectroscopy was used to determine the conductivity of LiBH_4_-Li_2_B_12_H_12_. Interestingly, in Figure 3c and Appendix A, it can be seen that the conductivity of LiBH_4_-Li_2_B_12_H_12_ (0.5 × 10^−6^ S cm^−1^ at RT) is improved compared to both pristine LiBH_4_ and Li_2_B_12_H_12_ (2.5 × 10^−8^ S cm^−1^ at RT) [55]. The improvement in ionic conductivity can be attributed again to a partial LiBH_4_ decomposition, to form some amorphous Li_2_B_12_H_12_ or [Li_2_B_12_H_11+1/n_]_n_ and undecomposed LiBH_4_ (see also next paragraph). Consequently, the improved cation mobility observed in the LiBH_4_-LiBF_4_ mixture prepared at 280 °C can be attributed to the formation of Li_2_B_12_H_12_ and its interaction with residual LiBH_4_, while the presence of LiF likely has a minor or no impact.

Table 1 shows the values of the activation energies (E_a_) obtained by a linear fit of the Arrhenius plot. The E_a_ of the orthorhombic LiBH_4_ is in good agreement with the calculated average value using all values reported in the literature by Gulino et al. [56], i.e., 0.75 ± 0.07 eV. The E_a_ obtained for the mixtures LiBH_4_-LiBF_4_ and LiBH_4_-Li_2_B_12_H_12_ are lower compared to that of pure LiBH_4_, indicating a more facile conduction mechanism, in agreement with the higher Li-ion conductivity. In addition, these values are in agreement with the data reported by Zhu et al. [38], e.g., 0.43 eV and 0.44 eV for the samples containing amorphous Li_2_B_12_H_12_ or [Li_2_B_12_H_11+1/n_]_n_, further indicating that the highly conductivity phase formed during reaction with LiBF_4_ could be amorphous Li_2_B_12_H_12_ or a related compound. It is also good to note that the oxidative stability of 4.0 V determined for the LiBH_4_-LiBF_4_ mixture (Appendix A) is similar to the value reported for other *closo*-borate-related compounds [13,38,57].

### 3.3. Alternative Decomposition Pathway

Although it seems clear that the formation of a Li_2_B_12_H_12_-like compound is responsible for the enhancement in ionic conductivity for LiBH_4_-LiBF_4_, this could not explain the behaviour of the system completely. Surprisingly, comparison of the respective conductivities of LiBH_4_-Li_2_B_12_H_12_ and LiBH_4_-LiBF_4_ mixtures (Figure 3c) illustrates that the latter has the highest conductivity. If the conductivity enhancement is solely related to the formation of Li_2_B_12_H_12_, this would not be the case. However, as our structural analysis did not yield any other compounds or phases, the unidentified conductive phase is likely very similar to amorphous Li_2_B_12_H_12_.

Interestingly, it has been proposed that the combination of different *closo*-borate complex anions could lead to improved ion transport in metal hydride-based ion conductors [38,54,58,59]. Toyama et al. prepared complex hydrides containing three *closo*-borate-type anions, e.g., [B_12_H_12_]^2−^, [B_11_H_11_]^2−^ and [B_10_H_10_]^2−^, from LiBH_4_ and B_10_H_14_ as the starting materials [54]. They observed that with increasing amount of [B_11_H_11_]^2−^ and [B_10_H_10_]^2−^, ionic conductivity improves. Remarkably, under the reaction conditions used during our synthesis, pyrolysis of B_2_H_6_ to B_10_H_14_ can occur [60,61]. The in-situ generated B_10_H_14_ can then react with LiBH_4_ to form several *closo*-borate anions, similar to the synthesis performed by Toyama et al. [54] The reaction scheme discussed here is presented in Figure 4. 

The conductivity enhancement achieved in decomposed LiBH_4_ in the presence of LiBF_4_ that forms multiple *closo*-borate phases would be higher than expected for a mixture consisting solely of LiBH_4_ and Li_2_B_12_H_12_. Consequently, the reaction process described in Figure 4 provides a possible explanation for an increased conductivity in LiBH_4_-LiBF_4_ compared to LiBH_4_-Li_2_B_12_H_12_ through the introduction of different lithium *closo*-borate compounds. A detailed investigation of this hypothesis will require solid-state ^11^B-MAS-NMR measurements, which is the subject of an ongoing study.

Another explanation for the enhanced conductivity could be the formation of highly conductive interfaces between the amorphous LiBH_4_-Li_2_B_12_H_12_ solid solution and the in-situ formed nanocrystalline LiF particles. The melting of the undecomposed LiBH_4_ will likely lead to strong interface interactions with LiF and Li_2_B_12_H_12_, thereby enhancing the ionic conductivity of the composite. Recent studies by Yan and Zhao reported on similar systems in which LiBH_4_ xNH_3_ was stabilized in an amorphous (molten-like) state on (inert) Li_2_O nanoparticles that were formed in-situ via a decomposition pathway [62,63]. Even in compounds with a high weight percentage (78 wt%) of the non-conducting metal oxide, a high room temperature conductivity has been obtained. Likewise, in our case, the amorphous LiBH_4_-Li_2_B_12_H_12_ phase could benefit from the presence of LiF particles, for example by stabilizing the amorphous form or even by beneficial interface interactions, thereby forming a conductive space-charge layer.

## 4. Conclusions

In conclusion, the impact of LiBF_4_ addition on the ionic transport in LiBH_4_ was explored by correlating the physical properties and ionic conductivity of LiBH_4_-LiBF_4_ mixtures. After synthesis at 280 °C, the LiBH_4_-LiBF_4_ sample contains LiF and amorphous lithium *closo*-borate, such as Li_2_B_12_H_12_. The presence of this phase is in line with the fact that partial substitution of the BH_4_ anion in LiBH_4_ with BF_4_ anion yields unstable LiBH_3_F, which decomposes into LiF and B_2_H_6_. The latter may react further to form B_10_H_14_, which in turn leads to the formation of [B_12_H_12_]^2−^, [B_11_H_11_]^2−^ and [B_10_H_10_]^2−^ complex anions by reacting with excess LiBH_4_.

Interestingly, a conductivity enhancement of over two orders of magnitude (to 0.9 × 10^−5^ S cm^−1^ at 30 °C) is achieved for the LiBH_4_-LiBF_4_ mixture reacted at 280 °C. The formation of LiF does not seem to lead to any improvement in conductivity, while the presence of solely Li_2_B_12_H_12_ leads to a minor enhancement. Thus, the improved lithium-ion mobility in LiBH_4_ through decomposition reaction with LiBF_4_ may be attributed to the formation of a lithium *closo*-borate-like phase, i.e., amorphous Li_2_B_12_H_12_ or [Li_2_B_12_H_11+1/n_]_n_. The insights provided in this work highlight the versatility of LiBH_4_ as a promising solid-state electrolyte, considering that several different strategies can be exploited to increase the ionic conductivity of the compound. Our work demonstrates that decomposition reactions is a promising approach to enhancing ionic conductivity in LiBH_4_. This approach might be applicable to other complex hydrides and other classes of inorganic solid electrolytes. 

## Figures and Tables

**Figure 1 molecules-27-02187-f001:**
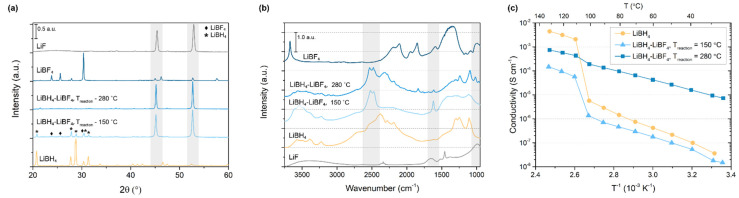
(**a**) X-ray powder diffraction patterns, (**b**) DRIFTS spectra and (**c**) Arrhenius conductivity plots of LiBH_4_-LiBF_4_ mixtures prepared at 150 °C and 280 °C. For comparison, LiBH_4_, LiBF_4_ and LiF are included in the XRD patterns; the spectra of LiBH_4_, LiBF_4_ and LiF are shown in the DRIFTS spectra and the ionic conductivity of pristine LiBH_4_ is added to the Arrhenius plots. Relevant reflections are highlighted in grey.

**Figure 2 molecules-27-02187-f002:**
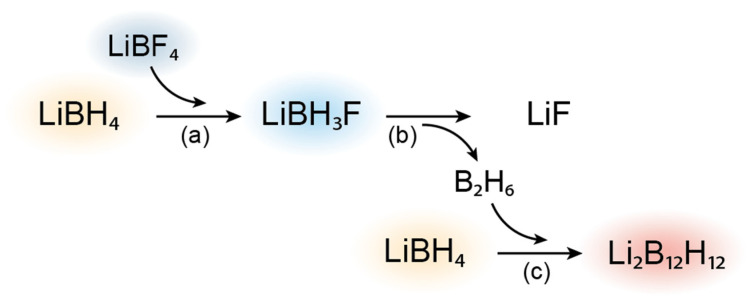
Reaction overview for LiBH_4_-LiBF_4_ solid solutions as proposed by Richter et al. [36]. (**a**) Hydrogen-fluorine exchange of LiBF_4_ with LiBH_4_ to form LiBH_3_F. (**b**) Decomposition of the substituted material yielding LiF and B_2_H_6_. (**c**) Reaction of B_2_H_6_ with excess LiBH_4_ forming Li_2_B_12_H_12_. Reaction stoichiometry not taken in account in this overview.

**Figure 3 molecules-27-02187-f003:**
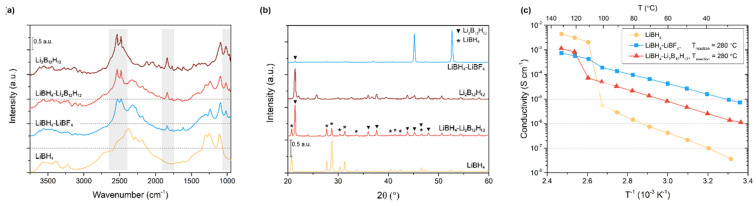
(**a**) DRIFTS spectra, (**b**) XRD diffraction patterns and (**c**) Arrhenius conductivity plots of LiBH_4_-LiBF_4_ and LiBH_4_-Li_2_B_12_H_12_ mixtures synthetized at 280 °C. In the DRIFTS and XRD graphs, LiBH_4_ and Li_2_B_12_H_12_ are included for comparison. In the DRIFTS, spectra peaks related to Li_2_B_12_H_12_ vibrations are highlighted in grey.

**Figure 4 molecules-27-02187-f004:**
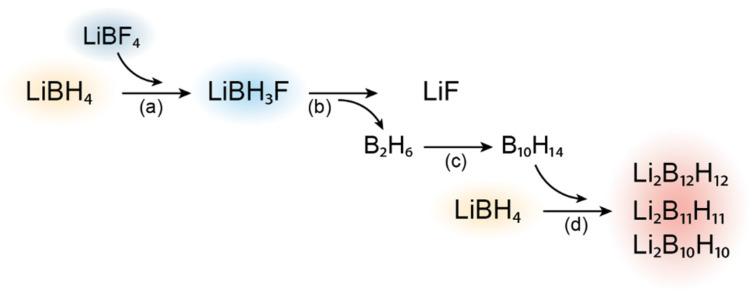
Reaction scheme proposed for the synthesis of LiBH_4_-LiBF_4_ solid solutions constituted of multiple *closo*-borate phases. (**a**) Reaction between LiBF_4_ and LiBH_4_ yielding unstable LiBH_3_F followed by (**b**) decomposition to form LiF and B_2_H_6_. (**c**) Pyrolysis of B_2_H_6_ generating B_10_H_14_ in-situ. (**d**) Reaction of B_10_H_14_ with LiBH_4_ forming several lithium *closo*-borates. Reaction stoichiometry not taken in account in this overview.

**Table 1 molecules-27-02187-t001:** Activation energies (eV) calculated from a linear plot of ln(σT) and 1000/T of the EIS data reported in Figure 3c. The standard deviation is based on the 99% confidence interval of the linear fit.

Sample	E_a_ (eV)	Pre-Exponential Factor
LiBH4 (orthorhombic)	0.68 (±0.05)	15 (±2)
LiBH4-LiBF4 (280 °C)	0.44 (±0.01)	11.1 (±0.3)
LiBH4-Li2B12H12 (280 °C)	0.41 (±0.01)	11.8 (±0.4)

## Data Availability

The data presented in this study are available in Appendix A.

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
