# Peer review of "Effects of LiBF4 Addition on the Lithium-Ion Conductivity of LiBH4"

_molecules, 2022, doi:10.3390/molecules27072187_

Round 1
Reviewer 1 Report
The proposed manuscript concerns an interesting topic of preparation and characterization a new solid-state electrolyte for solid-state batteries. I believe this subject is compatible with the journal scope. At the same time the manuscript needs some improvements to be considered for publication.
1. Page 1, lines 26-28: “The performance and safety requirements for automotive electrification and grid-energy storage systems, however, cannot be met by the commercially available batteries.” - it is advisable at this point that the authors refer to a review publication from recent years, for example: 10.3390/en14092649 (2021), 10.1016/j.susmat.2021.e00297 (2021)
2. Page 3, lines 118-128: Which software was used to simulate the EIS spectra?
3. Page 5, Figure 2.: Please insert a reference in the figure description after “...proposed by Richter et al.”. It will probably be [38].
4. Page 7, line 284: There is only one table in the main manuscript – please correct the table numbering.
5. Supporting information, S2, Figure S1: The figure seems to show the wrong direction of frequency – please explain.
Were Bode diagrams also created for the studied systems?
Whether parameter values for replacement circuits have been calculated?
Reviewer 2 Report
This manuscript investigated the effect of LiBF4 addition on enhancing the ionic conductivity of LiBH4 conductor. The IR spectroscopy, XRD patterns and electrochemical impedance spectroscopy for ionic conductivity tests confirm that conductivity enhancement of LiBH4-LiBF4 system originates from the formation of lithium closo-borate-like phase (i.e. amorphous Li2B12H12 or [Li2B12H11+1/n]n). Overall, this work will be of great interest for the materials and battery communities. Moreover, this manuscript is well organized and the results are clearly presented. Thus, I would like to recommend its publication after addressing the following concerns.
(1) Could the authors provide the voltage window of this LiBH4-LiBF4 mixture? Are the formed closo-borate phases stable when the voltage is above 4V vs.Li/Li+?
(2) Is it possible to show some electrochemical data of LiBH4-LiBF4 solid electrolytes for Li-metal batteries?
(3) Could the authors add the XRD pattern of LiF in Figure 1a?
